# Biochemical, Hematological, Inflammatory, and Gut Permeability Biomarkers in Patients with Alcohol Withdrawal Syndrome with and without Delirium Tremens

**DOI:** 10.3390/jcm13102776

**Published:** 2024-05-08

**Authors:** Mark M. Melamud, Daria V. Bobrik, Polina I. Brit, Ilia S. Efremov, Valentina N. Buneva, Georgy A. Nevinsky, Elvina A. Akhmetova, Azat R. Asadullin, Evgeny A. Ermakov

**Affiliations:** 1Institute of Chemical Biology and Fundamental Medicine, Siberian Branch of the Russian Academy of Sciences, 630090 Novosibirsk, Russia; marken94@mail.ru (M.M.M.); buneva@niboch.nsc.ru (V.N.B.); nevinsky@niboch.nsc.ru (G.A.N.); 2Department of Psychiatry and Addiction, Bashkir State Medical University, 450008 Ufa, Russia; dbobrik@internet.ru (D.V.B.); aea1202@yandex.ru (E.A.A.); droar@yandex.ru (A.R.A.); 3Department of Natural Sciences, Novosibirsk State University, 630090 Novosibirsk, Russia; 4Institute of Personalized Psychiatry and Neurology, Shared Core Facilities, V.M. Bekhterev National Medical Research Centre for Psychiatry and Neurology, 192019 Saint Petersburg, Russia; efremovilya102@gmail.com

**Keywords:** alcohol use disorder, alcohol withdrawal syndrome, Delirium Tremens, blood test, biochemical test, inflammation, gut permeability, FABP2, LBP, zonulin

## Abstract

**Background:** Delirium Tremens (DT) is known to be a serious complication of alcohol withdrawal syndrome (AWS). Neurotransmitter abnormalities, inflammation, and increased permeability are associated with the pathogenesis of AWS and DT. However, the biomarkers of these conditions are still poorly understood. **Methods:** In this work, biochemical, hematologic, inflammatory, and gut permeability biomarkers were investigated in the following three groups: healthy controls (n = 75), severe AWS patients with DT (n = 28), and mild/moderate AWS without DT (n = 97). Blood sampling was performed after resolution of the acute condition (on 5 ± 1 day after admission) to collect clinical information from patients and to investigate associations with clinical scales. Biomarker analysis was performed using automated analyzers and ELISA. Inflammatory biomarkers included the erythrocyte sedimentation rate (ESR), high-sensitivity C-reactive protein (hsCRP), and platelet-to-lymphocyte ratio (PLR). **Results:** Among the biochemical biomarkers, only glucose, total cholesterol, and alanine aminotransferase (ALT) changed significantly in the analyzed groups. A multiple regression analysis showed that age and ALT were independent predictors of the CIWA-Ar score. Hematologic biomarker analysis showed an increased white blood cell count, and the elevated size and greater size variability of red blood cells and platelets (MCV, RDWc, and PDWc) in two groups of patients. Gut permeability biomarkers (FABP2, LBP, and zonulin) did not change, but were associated with comorbid pathologies (alcohol liver disease and pancreatitis). The increase in inflammatory biomarkers (ESR and PLR) was more evident in AWS patients with DT. Cluster analysis confirmed the existence of a subgroup of patients with evidence of high inflammation, and such a subgroup was more frequent in DT patients. **Conclusions:** These findings contribute to the understanding of biomarker variability in AWS patients with and without DT and support the heterogeneity of patients by the level of inflammation.

## 1. Introduction

Alcohol use disorder (AUD) is one of the most urgent medical and social problems worldwide. According to a World Health Organization report, 8.7% of adult males and 1.7% of adult females suffered from AUD in 2016 [1]. In 2020, alcohol consumption was responsible for 1.78 million deaths worldwide and was the leading cause of death in men aged 15–49 [2]. More than 5% of all deaths occur due to alcohol abuse. In addition, alcohol use disorders also account for more than 5% of the global burden of disease [3].

While alcohol abuse is an obvious problem, abruptly stopping drinking alcohol can also be dangerous. Alcohol withdrawal syndrome (AWS) develops after cessation of alcohol intake after prolonged use. AWS is a condition with varied clinical manifestations that resolves within hours or days after cessation of alcohol consumption. The spectrum of clinical manifestations of AWS ranges from anxiety, sweating, and dizziness to severe conditions including Delirium Tremens (DT), alcoholic hallucinosis, alcoholic delusional psychosis, and alcoholic encephalopathies [4,5,6]. DT is a dangerous manifestation of AWS sometimes leading to the death of the patient. DT usually occurs three days after alcohol withdrawal and lasts several days. DT is accompanied by both somatic (tremors, heart rhythm disturbances, increased sweating) and mental (hallucinations, delusions) symptoms. Sometimes, people with DT experience the most dangerous symptoms, fever and seizures, which can lead to death [7]. Among people hospitalized with AUD, DT develops in 3–5% of patients. The mortality rate for DT without treatment can be as high as 37% [8].

The pathogenesis of AWS is known to be associated with a deficiency of inhibitory neurotransmitters (primarily gamma-aminobutyric acid) and the accumulation of dopamine [4,7]. However, recent evidence suggests the involvement of inflammation in the pathogenesis of alcohol dependence and withdrawal syndrome [9,10]. Increased concentrations of inflammatory biomarkers have been described in both AUD [11] and AWS with and without DT [12]. The number of lymphocytes is known to increase in the blood for both short-term and long-term alcohol consumption. It has been shown that healthy people have a significant increase in the number of leukocytes in the blood after drinking alcohol for 2–4 h [13]. There is also evidence that people who abuse large doses of alcohol for a long time also have elevated white blood cells in the blood [14]. In addition, patients with a long history of alcohol abuse are known to have increased concentrations of conventional inflammatory markers such as IL-6 [15] or TNF-α [11,16], with some cytokines like IL-8 and MCP-1 having a direct correlation with the amount of alcohol consumed per day [17]. AWS is also associated with changes in biochemical biomarkers, including liver enzyme activity [18,19]. Thus, hematological, biochemical, and inflammatory biomarkers change dynamically during AWS. However, the available data on the clinical relevance of these biomarkers are insufficient.

Impaired intestinal barrier function and bacterial translocation play an important role in the development of inflammation in AUD [20]. These processes are associated with increased circulation of pathogen-associated molecular patterns (PAMPs), including lipopolysaccharide (LPS). PAMPs, in turn, trigger an inflammatory response through conserved pattern recognition receptors (PRRs). It has been shown that LPS levels in the blood of patients with AUD are increased [21], and lipopolysaccharide binding protein (LBP) concentrations are increased in AWS [22]. In addition, there is evidence of increased TLR4 (recognizing LPS) activation in AUD patients [23] and increased TLR4 gene expression levels in alcoholized rats [24].

There are few studies describing changes in hematological and inflammatory markers in DT patients. A recent 2023 paper showed that leukocyte and neutrophil counts, and neutrophil-to-lymphocyte or platelet-to-lymphocyte ratios, are increased in AWS patients with DT compared to patients without DT [12]. Interestingly, lymphocyte and platelet levels are conversely significantly lower in DT patients [12]. Silczuk et al. also indicate that thrombocytopenia develops in patients with DT [25]. To our knowledge, there are currently no studies regarding markers of gut permeability in DT.

The main aim of this work was to investigate biochemical, hematological, inflammatory, and gut permeability biomarkers in AWS patients depending on the presence of DT and in comparison with healthy controls. In this work, it was planned to identify the biomarkers characteristic of the more severe course of AWS, i.e., delirium. The working hypothesis was that the pathogenesis of AWS and DT is partly related to inflammation caused by increased gut permeability. In contrast to existing studies investigating acute changes in biomarkers (blood sampling on admission), in this study, blood sampling for biomarker analysis was performed after the alleviation of AWS symptoms (on the fifth day after admission). Given the dynamic nature of changes in the levels of many biomarkers, this approach allows for the detection of long-lasting and more persistent abnormalities rather than transient changes in different biomarkers. This approach also provided an opportunity to examine the association of biomarkers with clinical scales assessing the severity of AWS, depressive symptoms, insomnia, and aggression, which are difficult to obtain at patient admission due to the severity of the condition.

## 2. Materials and Methods

### 2.1. Participants

This study was approved by the Local Ethics Committee of the ICBFM SB RAS (the protocol N3 from 19 June 2023) and conducted in accordance with the Declaration of Helsinki (2013 revision) [26]. Recruitment of participants was organized at the Republican Narcological Dispensary of the Ministry of Health of the Republic of Bashkortostan (Russian Federation). All participants agreed to participate in the study and signed informed consent. From June 2023 to January 2024, 125 patients with AWS and 75 healthy controls were included in the study.

The main inclusion criteria for patient were as follows: (1) diagnosis according to the International Classification of Diseases 10th Revision (ICD-10) Version 2016: F10.3 (Withdrawal state) or F10.4 (Withdrawal state with delirium); (2) age 18–60 years; (3) consent to participate in the study. In addition, only males were included in the study because males are more likely to be hospitalized with these conditions.

Exclusion criteria for patients were as follows: (1) severe, decompensated, or unstable comorbid somatic diseases, traumatic brain injury, alcoholic hallucinosis, alcoholic delusional psychosis, alcoholic encephalopathy, HIV infection, or viral hepatitis; (2) concomitant mental illnesses (according to ICD-10: schizophrenia—F20–F29, epilepsy—G40); (3) mental and behavioral disorders associated with the use of psychoactive substances (F10.1, F11.0−F19.0), dependence on psychoactive substances (except for nicotine and caffeine); (4) history of autoimmune diseases and cancer; (5) acute infectious diseases one month before the study; (6) acute allergic reactions one month before the study; (7) operations and blood transfusions one month before the study; (8) vaccination against any infections within a month before blood collection. The exclusion of the above pathologies is explained by the consideration that many of them are accompanied by an inflammatory response and therefore can significantly affect the analyzed biomarkers. Exclusion of patients with other types of dependencies was necessary to focus only on AUD.

The inclusion criteria for healthy controls were as follows: (1) absence of alcohol abuse and somatic diseases according to the results of a questionnaire before inclusion in the study; (2) age 18–60 years; (3) consent to participate in the study.

Recruited patients with AWS were then divided into 2 subgroups: (1) severe AWS patients with DT (F10.4) and (2) patients with mild/moderate AWS (without DT).

### 2.2. Sample Collection

Blood sampling was carried out on 5 ± 1 day after admission, that is, after resolution of AWS and relief of the acute condition. One of the objectives of the study was related to the analysis of associations of biomarkers with various clinical scales, but at the time of admission, it was impossible to conduct a clinical assessment by interviewing patients, so blood sampling and clinical characteristics of patients were carried out after normalization of the condition.

Blood samples were collected in four different BD Vacutainer blood collection tubes (BD, Franklin Lakes, NJ, USA). Three tubes were used for biochemical and hematological analysis. The remaining tube was used to obtain plasma and subsequent gut permeability biomarkers analysis. Plasma was extracted by centrifugation for 30 min at 2000× *g* and 4 °C. Plasma samples were stored at −80 °C until analysis.

### 2.3. Clinical Assessment

The clinical status of patients was assessed by experienced clinicians. A variety of clinical and anamnestic data were collected: age, age of onset of episodic and systematic use, duration of addiction, age of formation of AWS, AWS duration, maximum daily tolerance, and body mass index (BMI).

The following scales were used for clinical assessment of patients:AUDIT—Alcohol Use Disorders Identification Test [27];CIWA-Ar—Clinical Institute Withdrawal Assessment for Alcohol Scale [28];CGI-S—Clinical global impression—severity scale [29];MADRS—Montgomery–Asberg Depression Rating Scale [30];HDRS—Hamilton Rating Scale for Depression [31];ESS—Epworth Sleepiness Scale [32];ISI—Insomnia Severity Index [33];BPAQ24—Buss–Perry Aggression Questionnaire [34].

The presence of comorbid somatic pathology was also carefully considered. Comorbid diseases were diagnosed in accordance with international recommendations based on physical examination methods and laboratory tests. The presence of alcoholic liver disease (ALD (K70)), alcohol-induced chronic pancreatitis (K86), and arterial hypertension (I10–I15) was considered in this work. Among patients with ALD, there were mainly those diagnosed with alcoholic fatty liver (K70.0) and alcoholic hepatitis (K70.1). Patients with alcoholic hepatic failure (K70.4) were excluded.

### 2.4. Laboratory Measurements

Biochemical analysis was carried out using a BioSystems A-15 automatic analyzer (BioSystems S.A., Barcelona, Spain). Data on the concentrations of glucose, cholesterol, triglycerides, total protein, albumin, total bilirubin, creatinine, urea, uric acid, and K^+^ and Na^+^ ions were obtained using this analyzer. Enzyme activities (alanine aminotransferase (ALT), amylase, aspartate aminotransferase (AST), gamma-glutamyl transferase (GGT), creatine phosphokinase-MB (CPK-MB), and alkaline phosphatase (ALP)) were also determined using this analyzer. A BioSystems A-15 automatic analyzer was also used to analyze the following inflammatory biomarkers: the erythrocyte sedimentation rate (ESR) and high-sensitivity C-reactive protein (hsCRP).

Hematological analysis was performed using an automatic analyzer Abacus Junior 380 (Diatron MI Zrt., Budapest, Hungary). Using the analyzer, the following data were obtained: content of white blood cells (WBCs), lymphocytes (LYMs), red blood cells (RBCs) and platelets (PLTs), hemoglobin concentration (HGB), hematocrit (HCT), mean corpuscular volume of red blood cells (MCV), mean corpuscular hemoglobin (MCH), mean corpuscular hemoglobin concentration (MCHC), red cell distribution width (RDWc), plateletcrit (PCT), mean platelet volume (MPV), and platelet distribution width (PDWc). However, it did not allow for the determination of subpopulations of leukocytes.

Several indices were calculated based on the data obtained:De Ritis ratio (AST/ALT ratio) is a laboratory marker of liver damage including alcoholic liver disease [35,36];AST to Platelet Ratio Index (APRI) is a valuable marker of fibrosis and cirrhosis [37];Platelet-to-lymphocyte ratio (PLR) is a marker of platelet aggregation and systemic inflammation [38].

### 2.5. Analysis of Gut Permeability Biomarkers

FABP2, LBP, and zonulin concentrations were analyzed by ELISA in plasma. The following kits were used:(1)ELISA Kit for Fatty Acid Binding Protein 2, Intestinal (FABP2) (Cat. # SEA559Hu, Cloud-Clone Corp., Wuhan, China);(2)ELISA Kit for Lipopolysaccharide Binding Protein (LBP) (Cat. # SEB406Hu, Cloud-Clone Corp., Wuhan, China);(3)Human zonulin ELISA kit zonulin 1000 ng/mL Competitive ELISA (Cat. # E01Z0004, BlueGene Biotech Co., Ltd., Shanghai, China).

All stages of the analysis were carried out in accordance with the manufacturer’s recommendations. The results were recorded at a wavelength of 450 nm on a Multiskan FC Microplate Photometer (Thermo Fisher Scientific GmbH, Dreieirch, Germany).

The sensitivity (minimum detectable concentration) of the FABP2, LBP, and zonulin kits was as follows: 0.061, 0.67, and 1 ng/mL, respectively. The FABP2 concentration in some samples was below the minimum detectable concentration and was beyond the lower limit of the calibration curve, so such samples were considered FABP2 negative. Samples with concentrations above 0.061 ng/mL were considered FABP2 positive. The detected concentrations of LBP and zonulin were within the calibration curve.

### 2.6. Statistical Analysis

STATISTICA 10 (StatSoft. Inc., Tulsa, OK, USA) and OriginPro 2021 (OriginLab, Northampton, MA, USA) were used to perform the statistical analysis. The Shapiro–Wilk test was used to test the normality of distribution of each quantitative variable. Most of the variables had a non-normal distribution, so the data were presented as median (Q1, Q3) and nonparametric tests were used. The significance of the differences in biomarker levels was calculated using the Kruskal–Wallis test with Dunn’s post hoc test for multiple comparisons, because more than two groups were analyzed. The Mann–Whitney *U* test was used in case of comparison of two groups. Pearson’s chi-squared test was used to analyze the significance of the differences in categorical variables (including the ratio of FABP2-positive and negative samples in groups). The Spearman rank correlation coefficient was calculated to evaluate the correlation dependences. A multiple regression analysis was carried out in STATISTICA 10 (StatSoft. Inc., Tulsa, OK, USA), considering the CIWA-Ar score as the dependent variable and age, PLR, and ALT as the independent variables. A *p*-value of less than 0.05 was considered to be statistically significant. K-means clustering was used to classify patients and healthy individuals into two clusters/subgroups with signs of high and low inflammation, as previously described [39]. Clustering was based on data from the following inflammatory biomarkers: WBC count, PLT count, ESR, and PLR. The hsCRP concentration data were not used for clustering because they were not available for healthy donors. Graphs were generated using OriginPro 2021 (OriginLab, Northampton, MA, USA).

## 3. Results

### 3.1. Clinical Characteristics of the Study Groups

Three groups were formed after recruitment in this study: a group of healthy controls (HC—75 individuals), a group of patients with mild/moderate AWS (AWS without DT—97 individuals), and a group with severe AWS (AWS with DT—28 individuals). Clinical data of the groups are presented in Table 1. Only males were included in the study. The age of AWS patients with and without DT did not differ, but the HC group was younger than the AWS group of patients without DT. Otherwise, the HC group did not differ from the AWS groups. AUDIT scores for AWS patients indicate harmful alcohol use, which is expected. However, there were no differences in AUDIT scores between patients with and without DT. Patients with DT had higher CIWA-Ar and CGI-S scores than patients without DT, which is also understandable. Depressive symptoms assessed using the MADRS and HDRS did not differ between the two patient groups. The patients also did not differ in the level of sleepiness, which did not exceed the norm of 10 points, according to ESS scores. Patients suffered from moderate insomnia but sleep disturbances did not differ significantly between patient groups. The level of aggression between groups of patients according to BPAQ 24 also did not differ.

### 3.2. Biochemical Biomarkers in AWS

The biochemical data for the three groups analyzed are presented in Table 2.

Analysis of biochemical data showed that only glucose and cholesterol levels changed in the AWS groups of patients compared to the HC group (Table 2). Blood glucose levels in healthy individuals and patients were within the normal range. However, patients with AWS without DT had significantly higher blood glucose levels than healthy donors. Patients with AWS with DT also had higher blood glucose levels, but this increase was not statistically significant (*p* = 0.1). Total blood cholesterol levels tended to decrease in AWS patients without DT (*p* = 0.09) and significantly decreased in AWS patients with DT (*p* = 0.003) compared to the HC group.

The remaining biochemical data were not available for the HC group, so further analysis was performed between groups of AWS patients. Among analyzed biomarkers, only ALT activity increased approximately one and sixth-tenths-fold in AWS patients with DT compared with those without DT. However, the AST/ALT ratio did not change significantly in patients with DT compared with patients without DT. The APRI index associated with fibrotic changes in the liver also did not change. Creatinine, urea, and electrolyte (K^+^ and Na^+^) levels were normal, indicating proper renal function and no dehydration.

### 3.3. Hematological Biomarkers in AWS

The hematologic analysis data are presented in Table 3. The increased WBC count and decreased percentage of lymphocytes in AWS patients compared to the HC group were the most striking changes in the hematologic picture. The WBC count increased significantly in both groups of patients, although they remained within the normal range. The decrease in the percentage of lymphocytes was associated with an increase in the WBC count, as lymphocytes count did not change. Interestingly, WBC counts did not differ between patients with and without DT. The observed increase in WBCs indicates a slight activation of inflammatory processes.

The indicators related to RBCs have also changed significantly (Table 3). The RBC count was significantly decreased in AWS patients with DT compared to the HC group and patients without DT. Changes in the RBC count are associated with changes in HCT, MCV, MCH, MCHC, and RDWc. MCV was similarly increased in both groups of patients, indicating macrocytosis. The increase in MCH in AWS patients with DT compared to healthy individuals was associated with a decrease in the RBC count, as the hemoglobin concentration did not change. MCHC was significantly decreased in both groups of patients. There was also an increase in RDWc in AWS patients with and without DT compared to healthy individuals, which was associated with an increase in MCV. Thus, RBCs in AWS patients were characterized by increased size and greater variability in size than in healthy individuals.

The PLT count did not change significantly between the analyzed groups (Table 3). However, an increase in MPV was detected in AWS patients without DT compared to controls. In addition, PDWc increased significantly in both patient groups. Indicators related to PLTs did not differ between patients with and without DT. Thus, there is an increase in the size and variability in the size of PLTs in AWS that may indicate bone marrow activation and abnormalities in PLT production.

### 3.4. Inflammatory Biomarkers in AWS

Results of the inflammatory biomarker analysis are presented in Table 4. ESR was significantly increased in AWS patients with DT compared to healthy controls and patients without DT. Data on hsCRP concentrations were available for patients only. The hsCRP level was slightly higher than normal and corresponded to indicators for low-grade inflammation. However, hsCRP did not differ between AWS patients with and without DT. PLR was significantly increased in patients with DT compared with patients without DT and the HC group. These data indicate the activation of inflammatory processes in AWS patients with the increase being more striking in patients with DT.

### 3.5. Gut Permeability Biomarkers in AWS

Activation of inflammatory response may be associated with increased gut permeability. Data on the biomarkers of gut permeability in AWS are presented in Figure 1. FABP2 was not detected in all samples by ELISA. Interestingly, FABP2 was detected significantly more often in patients than in healthy donors (*p* = 0.006) (Figure 1A). However, FABP2 concentrations in healthy subjects (Me [Q1, Q3]: 0.18 [0.11, 0.38]), patients without DT (0.18 [0.10, 0.29]), and with DT (0.23 [0.14, 0.32]) ng/mL) were not significantly different (Figure 1B). LBP levels also did not differ between healthy individuals, and patients without DT and with DT (3.1 [2.6, 4.0], 3.0 [2.5, 3.8], 3.4 [2.7, 4.0] ng/mL, respectively) (Figure 1C). Zonulin concentration in the blood of healthy individuals (158 [93, 295]), patients without DT (196 [97, 282]), and with DT (196 [104, 266] ng/mL) also did not differ (Figure 1D). Thus, the concentration of gut permeability biomarkers did not differ in the analyzed groups.

### 3.6. Associations with Comorbid Conditions

The observed changes in biochemical, hematologic, inflammatory, and gut permeability biomarkers may be associated with comorbid conditions. This work examined the association of biomarkers with ALD, chronic pancreatitis, and arterial hypertension. It is important to note that these comorbidities occurred with approximately the same frequency in AWS patients with and without DT (chi-square test, *p* > 0.05 in all cases). Specifically, ALD occurred in 15% of patients without DT and 21% with DT. The incidence of pancreatitis did not differ significantly and was 37% in the group of patients without DT and 50% in the group of patients with DT. Concomitant hypertension also occurred at similar rates (22% in patients without DT and 14% in patients with DT).

The presence of concomitant ALD in the overall patient group was associated with an increase in FABP2 and a decrease in LBP (Figure 2A,B). Analysis of changes in other biomarkers in patients with and without ALD is presented in Appendix A. Among other biomarkers, only the urea concentration was decreased in patients with ALD compared with patients without ALD (Appendix A).

Patients with pancreatitis had increased zonulin concentrations compared to patients without pancreatitis (Figure 2C). In addition, patients with pancreatitis had higher total triglyceride levels and were characterized by an earlier age of onset of systematic alcohol consumption (Appendix A).

Patients with arterial hypertension had higher PLR levels than patients without cardiovascular pathology (Figure 2D). The increase in PLR was associated with a significant increase in the PLT count in patients with arterial hypertension (Appendix A). Increased RDWc and decreased MCV were also found in patients with hypertension. Among the biochemical markers, the creatinine level was increased and the AST/ALT ratio was decreased in hypertensive patients. In addition, patients with hypertension were characterized by greater age than patients without hypertension (Appendix A).

### 3.7. Clinical Associations: Correlation and Multiple Regression Analysis

Correlation analysis of the studied biomarkers with clinical parameters and scales is presented in Figure 3. The complete correlation matrix can be found in Appendix A. Strong correlation among biomarkers was found for zonulin and LBP (Rs = –0.71, *p* = 1 × 10^−18^). It should be also noted that LBP was mainly negatively correlated with biochemical parameters, while zonulin was positively correlated (Appendix A). Zonulin was also negatively correlated with the age of the patients (Rs = −0.21).

Among the hematologic parameters, PDWc was positively correlated with the AUDIT score (Rs = 0.27), and RDWc with age (Rs = 0.31) and the age of AWS formation (Rs = 0.29).

Among biochemical biomarkers, the ALT/AST ratio was negatively correlated with BMI. Creatinine was positively correlated with BMI. Uric acid was negatively correlated with the CIWA-Ar score (Rs = −0.37). AWS duration was negatively correlated with total protein and albumin. Amylase was positively correlated with the age of AWS formation.

Among inflammatory biomarkers, PLR was positively correlated with the CIWA-Ar score (Rs = 0.2) and hsCRP was positively correlated with the ESS score (Rs = 0.27).

A multiple regression analysis considering the CIWA-Ar score as the dependent variable and age, PLR, and ALT as the independent variables was then performed (Table 5). These variables were selected because they showed correlations with the CIWA-Ar score. Despite some deviations from the normal distribution of variables, the number of cases was quite large (n = 102), so a multiple regression analysis is applicable in this case. Age (*p* = 0.035) and ALT (*p* = 0.005) were shown to be independent predictors of the CIWA-Ar score, while the PLT was not associated with CIWA-Ar score. Interestingly, age had a negative relationship and ALT had a positive relationship with the CIWA-Ar score (Table 5).

### 3.8. Cluster Analysis and Stratification of Patients by Level of Inflammation

Analyzed groups may differ in inflammation levels. Using K-mean cluster analysis, all participants were stratified by the level of inflammation (Figure 4). It was shown that, among healthy individuals, only 9% of participants were classified in the “high-inflammation” cluster. Among patients without DT, 34% of participants were classified with “high-inflammation”, and this increased to 57% in patients with DT. Differences in the percentage of healthy individuals and patients were significant (chi-square test), but differences between patients without and with DT did not reach statistical significance.

The results of the comparison of biomarker levels in AWS patients classified by inflammation level into the “low-” and “high-inflammation” clusters are presented in Appendix A. Expectedly, inflammatory biomarkers (WBC count, ESR, PLR) were higher in patients from the “high-inflammation” cluster. RDWc and PLT counts were also increased in the “high-inflammation” cluster. Among the clinical data, only the age of AWS formation was higher in patients from the “high-inflammation” cluster.

Taken together, the results of the cluster analysis indicate the heterogeneity of patients in terms of inflammation levels and the existence of subgroups with high levels of inflammation in AWS patients, with more of such patients among DT patients.

## 4. Discussion

### 4.1. Biochemical Biomarkers in AWS Patients with and without DT

Among the biochemical biomarkers, this study revealed a trend toward an increase in glucose levels in AWS patients with DT and a significant increase in patients without DT, although the concentrations were within the normal range (Table 2). According to the literature, AWS patients commonly experience hypoglycemia, which is associated with an unbalanced diet and liver and pancreatic damage [40]. It is likely that the observed increase in glucose levels in AWS patients is related to the lower age of the control group (Table 1). In addition, elevated fasting glucose levels may be associated with peripheral neuropathy in patients with AUD [41]. However, this study did not consider the presence of neuropathy in AWS patients. Additionally, hypoglycemia may depend on the stage of AWS. There is evidence that in AWS patients after alcohol withdrawal, blood glucose levels decreased slightly within a few days [42]. It is also known that in patients with alcohol withdrawal syndrome, blood glucose levels are related to the preferred type of alcohol. For example, vodka abusers are statistically significantly more likely to develop hypoglycemia than those who abuse beer or liquor [43]. Moreover, alcohol withdrawal may be associated with increased intake and craving for sugar in some patients [44]. There is also evidence that in severe AWS patients, blood glucose is slightly lower than in those who tolerate alcohol withdrawal more easily [45]. Thus, glucose levels can fluctuate widely depending on the stage of AWS and can be related to both alimentary causes and liver and pancreatic dysfunction.

The results of this study indicate a decrease in total cholesterol levels in AWS patients with DT compared with healthy individuals (Table 2). Literature data on cholesterol levels in AWS patients are inconsistent. There is evidence that total cholesterol levels are elevated in AWS patients compared to non-drinkers [46,47], which is associated with an unbalanced diet and liver damage [48]. However, there is evidence indicating that, in AWS, total cholesterol levels can be significantly reduced [47,49,50] and even become slightly lower than in healthy donors [47].

Among liver enzymes, this study showed an increase in ALT in patients with DT compared with patients without DT (Table 2). These results are supported by a meta-analysis published in 2014 [51]. The median ALT level in our study was approximately one and sixth-tenths times higher than normal. This result is consistent with another study showing increased ALT in AWS with delirium [52]. Wojnar et al. also considered ALT elevation as a predictor of DT [53]. There is also evidence that withdrawal is more severe in patients with more severe liver damage [52]. These findings may explain the higher ALT levels in AWS patients with DT.

In addition, it is important to consider the dynamic nature of liver enzyme changes. According to many studies, in the acute stage of AWS including accompanying DT, there is an increase in some liver enzymes, including ALT, AST, and GGT [54,55]. Since, in this study, blood was collected for analysis after resolution of severe symptoms, the findings indicate the normalization of many liver biomarkers. As stated above, only the elevation of ALT was observed. The other biomarkers were normal. Therefore, ALT abnormalities may be long-lasting and associated with liver damage.

### 4.2. Blood Biomarkers in AWS Patients with and without DT

This work shows an increase in the WBC count in both groups of patients compared to healthy individuals. This result is consistent with a recent study showing that patients with DT have significantly higher WBC counts than patients without DT [12]. The increased WBC found in this study may be associated with increased neutrophils. There is also evidence that patients with DT are characterized by a decreased lymphocyte level in peripheral blood [12]. In our study, the lymphocyte count also decreased, but did not reach the threshold of statistical significance (*p* = 0.16). An increased neutrophil count and decreased lymphocyte count and the neutrophil-to-lymphocyte ratio is considered a predictor of the development of DT [12]. These results support that a more severe course of AWS (accompanying DT) is associated with long-term WBC count changes.

The results obtained in this study about decreased RBC counts in patients with DT compared to patients without DT and healthy individuals are consistent with the literature data [56]. There is also evidence of increased MCV and MCH in people who abuse alcohol [55,57]. This study also showed an increase in these values compared to healthy donors. MCV values for patients were close to the values characteristic of macrocytosis. At the same time, MCV and MCH in patients with DT were, although not significantly, higher than in patients without DT. RDWc in the patient groups was also significantly higher than in healthy donors. An unbalanced nutrient-deficient diet characteristic of AUD patients may be associated with increased RDWc [58]. An increase in RDWc is considered an indicator of various forms of anemia. There is evidence that anemia is a fairly common condition in patients with alcohol dependence [59]. In addition, elevated RDWc levels may be associated with increased mortality risks in AUD patients [60]. Thus, the findings indicate impaired erythrocyte generation and maturation in the bone marrow with a shift toward increased cell size and distribution width.

The PLT count in all groups of patients and healthy individuals was within the normal range. According to the literature data, AWS patients have thrombocytopenia in the acute period [25,61]. The data obtained in this study indicate the recovery of the PLT count after the resolution of the acute condition, which is in agreement with the literature data [25,61]. Interestingly, PDWc in both groups of patients was more than twice the normal and statistically significantly different from the values of the group of healthy individuals. To date, there are quite few studies on PDWc changes in alcohol dependence found in the literature. Michalak et al. described an increase in this marker in patients with ALD compared to patients with non-alcoholic liver disease and healthy controls [62]. However, this study showed no differences in PDWc in patients with and without ALD (Appendix A). Nevertheless, PDWc can be considered as a marker of impaired PLT generation in bone marrow in patients with AWS.

Given the dynamic changes in hematologic parameters, the findings indicate that blood biomarkers are slower to normalize after the resolution of the acute condition than biochemical biomarkers.

### 4.3. Gut Permeability in AWS Patients with and without DT

Acute and chronic alcohol consumption is associated with changes in markers of intestinal permeability [63,64]. In this study, the levels of FABP2, LBP, and zonulin did not statistically significantly change in AWS patients compared to healthy individuals (Figure 1). Despite a trend toward increased FABP2 and LBP in patients with DT, the differences did not reach statistical significance. These results may be explained by insufficient sample power. Therefore, replicative studies on a larger sample of AWS patients, especially with DT, are needed. Another explanation for the lack of differences is related to variations in gut permeability biomarkers during the course of AWS. Increases in gut permeability biomarkers occur immediately during alcohol consumption and for a couple of days after alcohol withdrawal [20,22,65]. After some time of alcohol withdrawal, these biomarkers recovered to normal values [22,65]. Therefore, the results of this study may indicate the normalization of gut permeability biomarkers after the resolution of the acute condition. However, this assumption needs to be confirmed in subsequent longitudinal studies, especially with DT patients, as no such studies have been conducted in this patient population. This study also found that FABP2 was detected more frequently in AWS patients compared to healthy donors (*p* = 0.02). This can be considered as a residual phenomenon from intestinal damage after stabilization of the total biomarker level.

In addition, a negative correlation of LBP with biochemical parameters was found in this work. This correlation is related to the formation of LBP in the liver [66], so liver dysfunction promotes the decreased production of LBP and other proteins, and increased biochemical markers of liver damage. Zonulin is a tight contact protein and is synthesized outside the liver in many tissues, including the small intestine. Zonulin is involved in the disassembling of tight contacts in the intestinal epithelium [66]. An increase in zonulin can lead to an increase in gut permeability, which in turn can directly lead to impaired liver function and increased liver damage biomarkers. In turn, the strong negative correlation between LBP and zonulin (Rs = −0.71, Figure 3) is associated with similar processes. In this case, liver dysfunction caused by a zonulin-dependent increase in gut permeability leads to decreased synthetic liver function and decreased LBP formation.

### 4.4. Inflammation in AWS Patients with and without DT

The results of inflammatory biomarker studies indicate inflammation activation in AWS patients, especially with DT. It is shown that AWS patients with DT have an increased ESR and PLR compared to AWS patients without DT (Table 4). This result is consistent with previously published data [67]. PLR is also considered as an inflammatory marker. Increased PLR has also been shown in AWS patients with DT compared to patients without DT [12]. In addition, activation of inflammation in AUD and AWS has been shown in many studies [10,68,69,70].

The increase in inflammation is also confirmed by the results of cluster analysis (Figure 4). The proportion of participants with signs of high inflammation increased from 34% in patients without DT to 57% in patients with DT. These data support the existence of subgroups of patients with high inflammation in AWS and especially in DT. These findings may indicate the involvement of inflammation in the pathogenesis of AWS. Inflammation may worsen the severity of AWS and contribute to the development of DT.

These findings may have a translational perspective. The existence of a subgroup of patients with high inflammation dictates the need for anti-inflammatory therapy for these patients. The use of anti-inflammatory drugs may have a beneficial effect on a subset of these patients. Recently, there has been increased interest in the repurposing of drugs for the treatment of inflammatory and autoimmune diseases for the therapy of AUD and related conditions [71]. Such drugs with anti-inflammatory effects include ibudilast, apremilast, exenatide, semaglutide, liraglutide, metformin, and others, which have shown to reduce craving and the amount of alcohol drunk in animal models and randomized clinical trials in humans [71,72,73]. However, there is a need to develop criteria for stratifying patients by inflammation level for personalized anti-inflammatory therapy.

### 4.5. Associations with Comorbid Pathologies and Clinical Variables

Some biomarker changes may be associated with comorbid diseases. The findings of this study on the association of FABP2 and LBP with ALD (Figure 2) are supported by existing data [22,74]. The findings of increased zonulin in patients with concomitant pancreatitis are also consistent with the literature [75]. The observed increase in PLR in patients with arterial hypertension may be associated with the activation of inflammation and impaired PLT function [76]. Current evidence supports an association of PLR with arterial hypertension [77,78].

Significant correlations of biomarkers with clinical scales were also found in this study. ESS scores were positively correlated with hsCRP levels. This correlation may be associated with the effects of inflammation on excessive daytime sleep. The literature describes the association of inflammatory biomarkers with excessive daytime sleep and sleep disturbances [79,80,81]. AUDIT scores were correlated with several platelet indices. A positive correlation with PDWc is associated with liver dysfunction, as PDWc is known to be a biomarker of liver damage [62]. In addition, uric acid levels were negatively correlated with alcohol dependence duration and the CIWA-Ar score, which may be related to liver dysfunction. A decrease in the uric acid concentration in AWS has been described in the literature [82]. Decreased detoxifying function of the liver may affect the mental state of patients and is associated with the development of DT.

### 4.6. Limitations of the Present Study

This study has limitations. First, all biomarkers were examined at only one time point (on 5 ± 1 day after admission). A follow-up study from admission to discharge would have helped to understand much better the pattern of changes in the analyzed biomarkers in AWS patients. Secondly, the group of patients with DT was relatively small compared to the group of patients without DT and healthy individuals. Therefore, further studies on larger patient samples are required. Third, this study analyzed biomarker changes in men only. The all-male sample and the exclusion of certain demographics and comorbid conditions limited the generalizability of the results. Future studies should include not only men, but also women, and evaluate the effect of excluded conditions on the analyzed parameters. Fourth, the lack of significant differences in the level of gut permeability biomarkers may be related to insufficient study power. Therefore, further studies should include more participants to increase the study power. Last, not all biomarkers were available for comparison in healthy donors (for example, hsCRP), which was caused by insufficient sample volume for analysis. In addition, the study did not consider smoking status. Future studies should consider obtaining sufficient plasma for the study and possible confounding factors.

## 5. Conclusions

This work showed associations with changes in laboratory biomarkers after the resolution of acute symptoms in AWS patients with and without DT. The findings suggest an association of ALT with more severe AWS. Hematologic parameters indicate impaired RBC and platelet formation in the bone marrow. Gut permeability biomarkers have been associated with comorbid pathology (ALD and pancreatitis). Inflammatory biomarkers indicated higher levels of inflammation in patients with DT than in patients without DT. The results of cluster analysis confirm the existence of a subgroup of patients with evidence of high inflammation. There are more patients with high levels of inflammation among patients with DT. These findings support heterogenous inflammatory subgroups in AWS patients that may benefit from targeted anti-inflammatory interventions.

## Figures and Tables

**Figure 1 jcm-13-02776-f001:**
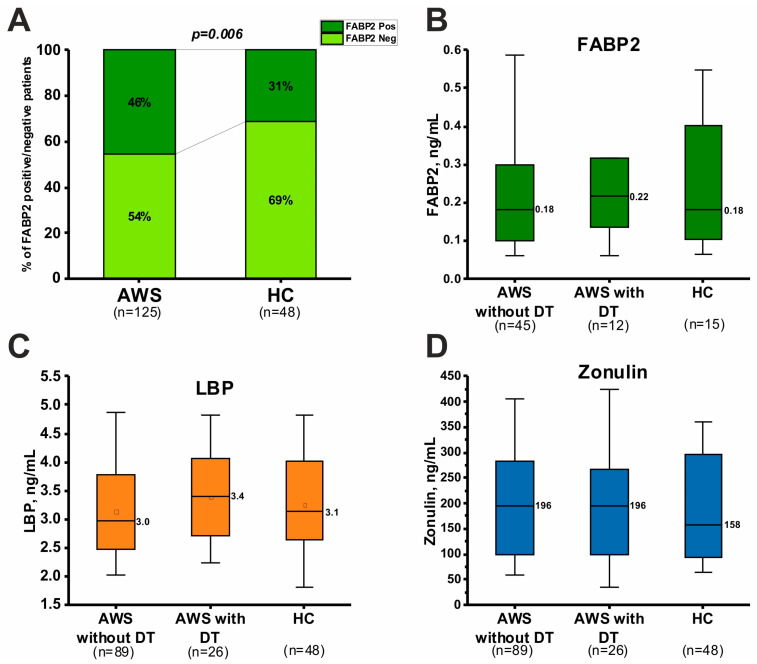
Plasma FABP2, LBP, and zonulin concentrations in patients with alcohol withdrawal syndrome (AWS) with and without Delirium Tremens (DT) compared with healthy controls (HCs). (**A**) Stacked histogram indicated the percentage of FABP2-positive and negative samples in the analyzed groups. The chi-square test was used to assess the significance of differences (*p*-value). (**B**–**D**) Plasma concentrations of FABP2 (**B**), LBP (**C**), and zonulin (**D**) in the analyzed groups. No significant differences were found using the Kruskal–Wallis test.

**Figure 2 jcm-13-02776-f002:**
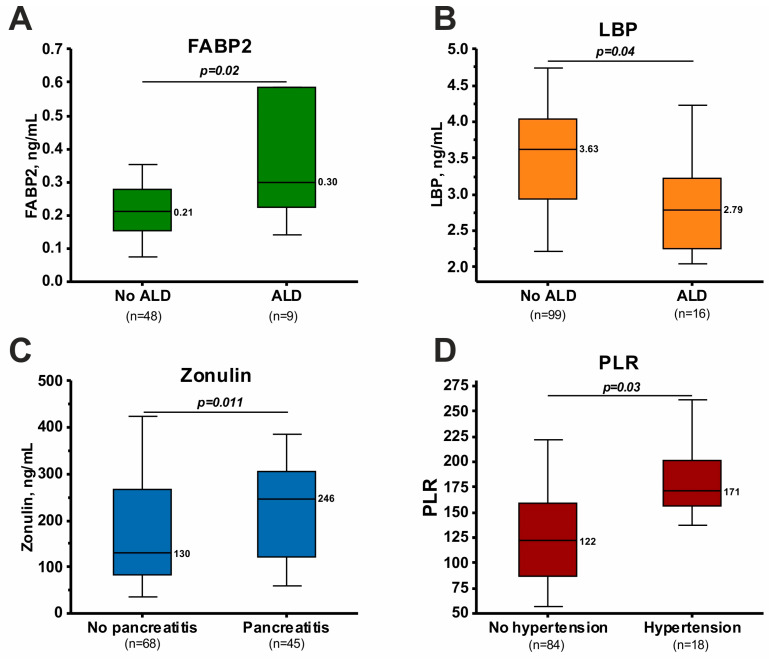
Dependence of biomarker levels on comorbid conditions in AWS. (**A**) FABP2 level in patients with and without ALD. (**B**) LBP level in patients with and without ALD. (**C**) Zonulin levels in patients with and without pancreatitis. (**D**) PLR in patients with and without hypertension. The differences were calculated using the Mann–Whitney test.

**Figure 3 jcm-13-02776-f003:**
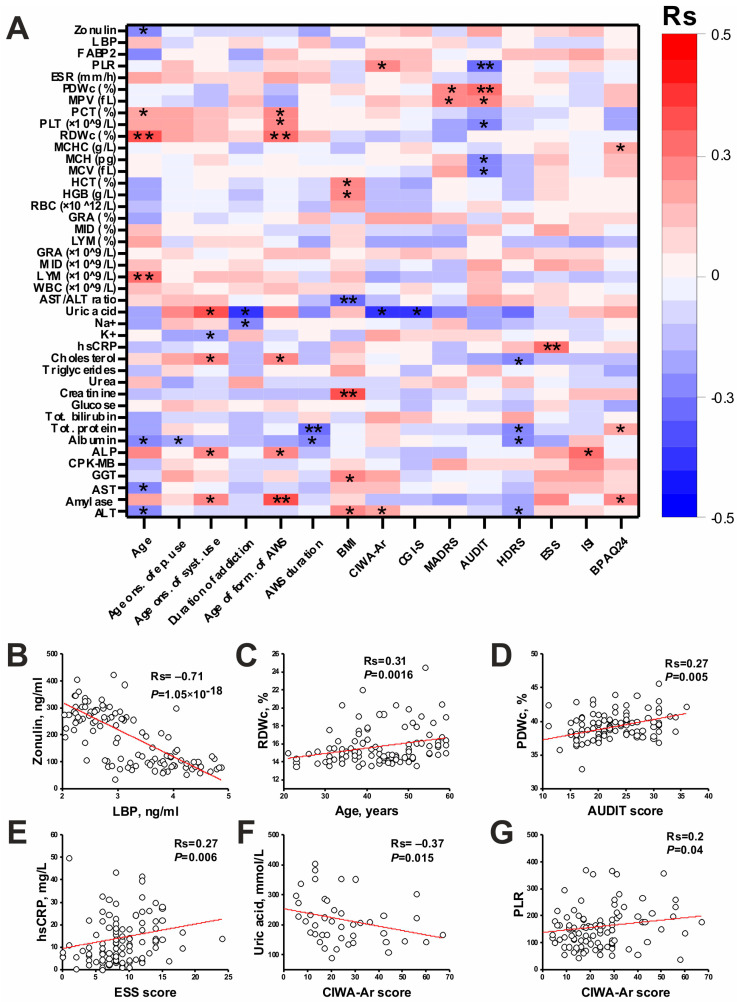
Correlation analysis of biochemical, hematological, inflammatory, and gut permeability biomarkers with clinical parameters and scales in a general group of patients. (**A**) Correlation heatmap (* *p* < 0.05, ** *p* < 0.01). (**B**–**G**) Scatterplots for the most significant correlations: zonulin and LBP (**B**), RDWc and age of patients (**C**), PDWc and AUDIT score (**D**), hsCRP and ESS score (**E**), uric acid and CIWA-Ar score (**F**), and PLR and CIWA-Ar score (**G**). Rs—Spearman’s correlation coefficient. For abbreviations, see Table 1, Table 2, Table 3 and Table 4.

**Figure 4 jcm-13-02776-f004:**
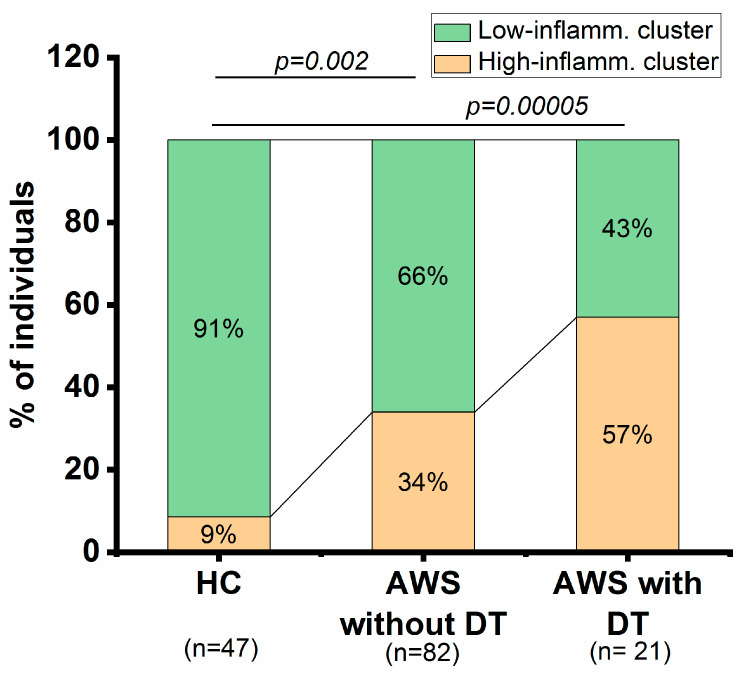
Stratification of patients with AWS according to the inflammation level using cluster analysis. Stacked histogram indicated the percentage of participants assigned to “low-inflammation” and “high-inflammation” clusters. Clustering was performed using the K-mean algorithm based on the following inflammatory biomarkers: WBC count, PLT count, ESR, and PLR. Differences were calculated using the chi-square test.

**Table 1 jcm-13-02776-t001:** Clinical data of participants included in the study.

Parameter	HC	n	AWS without DT	n	AWS with DT	n	*p*-Value (HC vs. AWS without DT)	*p*-Value (HC vs. AWS with DT)	*p*-Value (AWS with vs. without DT)
Age (years)	32.5 (23.8, 43.3)	75	45 (37, 51)	97	37 (34, 44.8)	28	**8.6 × 10^−7^**	0.36	0.11
Sex	Male	75	Male	97	Male	28	NA	NA	NA
Age of onset of episodic use (years)	NA	-	20 (17, 25)	95	20 (17, 24.5)	28	NA	NA	0.99
Age of onset of systematic use (years)	NA	-	24 (16, 29)	95	21.5 (18, 28.5)	28	NA	NA	0.23
Duration of addiction (years)	NA	-	11 (4.5, 20)	97	10.5 (2.3, 17.8)	28	NA	NA	0.46
Age of formation of AWS (years)	NA	-	31 (26, 39)	88	30 (25, 35)	28	NA	NA	0.24
Maximum daily tolerance, ml of abs. EtOH/kg	NA	-	6.2 (4.0, 9.1)	95	5.7 (4.5, 9.6)	28	NA	NA	0.73
AWS duration (days)	NA	-	3 (2, 3)	88	3 (2, 4)	28	NA	NA	0.12
BMI	23.3 (20.4, 26.2)	75	24.5 (22.5, 26.5)	95	22.9 (21.9, 24.4)	28	0.13	0.99	0.22
AUDIT (points)	NA	-	22 (19, 27)	95	22 (17.3, 27)	28	NA	NA	0.78
CIWA-Ar (points)	NA	-	18 (12.5, 24)	97	39 (20.3, 55.8)	28	NA	NA	**2.5 × 10^−7^**
CGI-S (points)	NA	-	4 (4, 5)	97	5 (5, 6)	28	NA	NA	**2.9 × 10^−5^**
MADRS (points)	NA	-	6 (2, 11.5)	97	8 (3.3, 13)	28	NA	NA	0.5
HDRS (points)	NA	-	10 (4.5, 14)	97	11 (4, 16.8)	28	NA	NA	0.53
ESS (points)	NA	-	8 (6, 12)	97	9 (7.3, 10)	28	NA	NA	0.99
ISI (points)	NA	-	12 (7, 14.5)	97	9 (7, 14)	28	NA	NA	0.36
BPAQ 24 (points)	NA	-	68 (54, 79.5)	97	68 (58.5, 76.5)	28	NA	NA	0.92

Note: Data presented as median (Q1, Q3). The significance of the differences was calculated using the Kruskal–Wallis test and the post hoc Dunn’s multiple comparisons test. Significant differences (*p* < 0.05) are highlighted in bold. Abbreviations: AWS—alcohol withdrawal syndrome, HCs—healthy controls, DT—Delirium Tremens, BMI—body mass index, NA—not applicable, AUDIT—Alcohol Use Disorders Identification Test, CIWA-Ar—Clinical Institute Withdrawal Assessment for Alcohol Scale, CGI-S—Clinical global impression—severity scale, MADRS—Montgomery–Asberg Depression Rating Scale, HDRS—Hamilton Rating Scale for Depression, ESS—Epworth Sleepiness Scale, ISI—Insomnia Severity Index, BPAQ24—Buss–Perry Aggression Questionnaire.

**Table 2 jcm-13-02776-t002:** Biochemical biomarkers in healthy individuals and AWS patients with and without DT.

Parameter	HC	n	AWS without DT	n	AWS with DT	n	*p*-Value (HC vs. AWS without DT)	*p*-Value (HC vs. AWS with DT)	*p*-Value (AWS with vs. without DT)
Glucose (mmol/L)	4.4 (4.2, 4.6)	46	5.1 (4.2, 6)	82	5 (4.1, 5.9)	22	**0.002**	0.1	0.99
Cholesterol (mmol/L)	5.1 (4.1, 5.8)	46	4.8 (4, 5.6)	82	4.3 (3.2, 5.3)	22	0.09	**0.003**	0.18
ALT (U/L)	ND	-	38 (21, 76)	82	62 (28, 114)	22	NA	NA	**0.04**
Amylase (U/L)	ND	-	60 (48, 71)	69	62 (40, 91)	17	NA	NA	0.94
AST (U/L)	ND	-	35 (24, 54)	82	47 (26, 96)	22	NA	NA	0.053
GGT (U/L)	ND	-	61 (26, 106)	82	87 (33, 217)	22	NA	NA	0.22
CPK-MB (U/L)	ND	-	19 (15, 25)	69	17 (12, 25)	17	NA	NA	0.43
ALP (U/L)	ND	-	160 (123, 194)	69	136 (104, 177)	17	NA	NA	0.09
Albumin (g/L)	ND	-	45 (42, 46)	82	43 (41, 46)	22	NA	NA	0.20
Total protein (g/L)	ND	-	73 (68, 79)	82	75 (70, 78)	22	NA	NA	0.56
Total bilirubin (umol/L)	ND	-	6.7 (4.6, 10.5)	82	9.6 (6, 11.5)	22	NA	NA	0.10
Creatinine (umol/L)	ND	-	78 (68, 96)	82	71 (53, 93)	22	NA	NA	0.14
Urea (mmol/L)	ND	-	3.9 (3.2, 4.6)	82	3.6 (3.2, 4)	22	NA	NA	0.71
Triglycerides (mmol/L)	ND	-	1.3 (0.9, 2.1)	82	1.4 (0.9, 2)	22	NA	NA	0.59
K^+^ (mmol/L)	ND	-	5 (4.6, 6.8)	81	5 (4.5, 5.5)	21	NA	NA	0.60
Na^+^ (mmol/L)	ND	-	144 (141, 145)	81	144 (141, 146)	22	NA	NA	0.97
Uric acid (mmol/L)	ND	-	219 (166, 283)	34	170 (138, 213)	10	NA	NA	0.10
AST/ALT ratio	ND	-	1.04 (0.78, 1.34)	82	1.1 (0.96, 1.4)	22	NA	NA	0.29
APRI	ND	-	0.32 (0.23, 0.70)	82	0.47 (0.24, 1.02)	21	NA	NA	0.38

Note: Data presented as median (Q1, Q3). The significance of the differences was calculated using the Kruskal–Wallis test and the post hoc Dunn’s multiple comparisons test. Significant differences (*p* < 0.05) are highlighted in bold. Abbreviations: AWS—alcohol withdrawal syndrome, HCs—healthy controls, DT—Delirium Tremens, NA—not applicable, ND—no data, ALT—alanine aminotransferase, AST—aspartate aminotransferase, GGT—gamma-glutamyl transferase, CPK-MB—creatine phosphokinase-MB, ALP—alkaline phosphatase, APRI—AST to Platelet Ratio Index.

**Table 3 jcm-13-02776-t003:** Hematological biomarkers in healthy individuals and AWS patients with and without DT.

Parameter ^1^	HC	n	AWS without DT	n	AWS with DT	n	*p*-Value (Kruskal–Wallis Test)	*p*-Value (HC vs. AWS without DT)	*p*-Value (HC vs. AWS with DT)	*p*-Value (AWS with vs. without DT)
WBCs (×10^9^/L)	5.2 (4.6, 5.8)	47	6.5 (5.4, 7.8)	82	6.5 (5.4, 9.1)	22	**1.4 × 10^−5^**	**3.7 × 10^−5^**	**9.9 × 10^−4^**	0.99
LYM (×10^9^/L)	1.7 (1.5, 2)	47	1.9 (1.5, 2.3)	82	1.5 (1.1, 2.1)	22	0.12	0.68	0.99	0.16
LYMs (%) ^2^	34.7 (29.2, 38.6); 33.95 ± 7.81	47	28.1 (22.7, 35.7); 28.8 ± 8.9	82	25 (15.4, 31.4); 24.8 ± 8.9	22	**3.2 × 10^−4^** **(1.6 × 10^−4^)**	**0.009** **(0.007)**	**5.4 × 10^−4^** **(5.5 × 10^−4^)**	0.23 (0.09)
RBCs (×10^12^/L)	4.2 (3.9, 4.5)	47	4.4 (4.1, 4.8)	82	4 (3.7, 4.3)	22	**9.1 × 10^−4^**	0.66	**6.2 × 10^−4^**	**0.007**
HGB (g/L)	131 (125, 140)	47	132 (121, 144)	82	125 (112, 136)	22	0.09	0.99	0.11	0.14
HCT (%) ^2^	40 (37, 41); 39.4 ± 3.3	47	42 (38, 46); 41.7 ± 5.1	82	39 (36, 43); 38.8 ± 4.8	22	**0.002** **(0.004)**	**0.007** **(0.002)**	0.99 (0.59)	**0.03** **(0.02)**
MCV (fL)	87 (85, 90)	47	96 (91, 100)	82	97 (93, 100)	22	**1.6 × 10^−11^**	**7.9 × 10^−11^**	**1.3 × 10^−6^**	0.99
MCH (pg)	29.4 (28.5, 30.5)	47	29.8 (28.6, 31.7)	82	31.4 (30, 32.4)	22	**0.008**	0.19	**0.006**	0.17
MCHC (g/L)	336 (333, 339)	47	315 (308, 320)	82	317.5 (312, 326)	22	**1.9 × 10^−19^**	**1.6 × 10^−19^**	**1.06 × 10^−7^**	0.89
RDWc (%)	13.2 (12.9, 13.7)	47	15.1 (14.5, 16.5)	82	14.9 (14.3, 16.4)	22	**3.5 × 10^−16^**	**8.6 × 10^−16^**	**9.8 × 10^−8^**	0.99
PLT (×10^9^/L)	238 (145, 276)	47	235.5 (171.5, 310.3)	82	290 (197, 478)	22	0.13	0.99	0.45	0.13
PCT (%)	0.25 (0.22, 0.3)	47	0.29 (0.23, 0.5)	82	0.26 (0.2, 0.34)	22	0.08	0.99	0.09	0.14
MPV (fL)	10.3 (9.1, 11.4)	47	11.1 (10.4, 12)	82	10.7 (10.1, 11.3)	22	**0.008**	**0.008**	0.99	0.47
PDWc (%)	15.9 (15.6, 16.1)	47	39.6 (38.1, 40.6)	82	38.5 (37.9, 40.3)	22	**4.7 × 10^−21^**	**6.6 × 10^−21^**	**8.9 × 10^−9^**	0.99

^1^ Data presented as median (Q1, Q3). The significance of the differences was calculated using the Kruskal–Wallis test and the post hoc Dunn’s multiple comparisons test. Significant differences (*p* < 0.05) are highlighted in bold. ^2^ These parameters had a normal distribution, therefore the mean ± standard deviation is also presented for them and the significance of differences between groups was additionally calculated using parametric tests (ANOVA and t-test). Abbreviations: AWS—alcohol withdrawal syndrome, HCs—healthy controls, DT—Delirium Tremens, WBCs—white blood cells, LYMs—lymphocytes, RBCs—red blood cells, HGB—hemoglobin, HCT—hematocrit test, MCV—mean corpuscular volume of red blood cells, MCH—mean corpuscular hemoglobin, MCHC—mean corpuscular hemoglobin concentration, RDWc—red cell distribution width, PLT—platelet count, PCT—plateletcrit, MPV—mean platelet volume, PDWc—platelet distribution width.

**Table 4 jcm-13-02776-t004:** Inflammatory biomarkers in healthy individuals and AWS patients with and without DT.

Parameter	HC	n	AWS without DT	n	AWS with DT	n	*p*-Value (Kruskal–Wallis Test)	*p*-Value (HC vs. AWS without DT)	*p*-Value (HC vs. AWS with DT)	*p*-Value (AWS with vs. without DT)
ESR (mm/h)	5.5 (4, 9.5)	47	6 (3, 15)	82	16 (4.5, 25)	22	**0.03**	0.99	**0.046**	**0.03**
hsCRP (mg/L)	ND	-	11.4 (4.9, 21.6)	82	10.2 (4, 17.1)	22	NA	NA	NA	0.46
PLR	140 (118, 178)	47	134 (86, 166)	82	213 (144, 274)	21	**7.3 × 10^−4^**	0.81	**0.02**	**4.3 × 10^−4^**

Note: Data presented as median (Q1, Q3). The significance of the differences was calculated using the Kruskal–Wallis test and the post hoc Dunn’s multiple comparisons test. Significant differences (*p* < 0.05) are highlighted in bold. Abbreviations: HCs—healthy controls, AWS—alcohol withdrawal syndrome, DT—Delirium Tremens, ND—no data, NA—not applicable, ESR—erythrocyte sedimentation rate, hsCRP—high-sensitivity C-reactive protein, PLR—platelet-to-lymphocyte ratio.

**Table 5 jcm-13-02776-t005:** Determinants of the CIWA-Ar score in multiple regression analysis, with age, PLR, and ALT as independent variables.

Variable	β	Standard Error of β	t Statistic	*p*-Value	Adjusted R^2^
Age	−0.294	0.138	−2.14	**0.035**	0.116
PLR	0.021	0.014	1.51	0.135	
ALT	0.066	0.023	2.87	**0.005**	

Note: Significant differences (*p* < 0.05) are highlighted in bold. Abbreviations: PLR—platelet-to-lymphocyte ratio, ALT—alanine aminotransferase.

## Data Availability

The data presented in this study are available from the corresponding author on reasonable request.

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
