# Peer review of "Biochemical, Hematological, Inflammatory, and Gut Permeability Biomarkers in Patients with Alcohol Withdrawal Syndrome with and without Delirium Tremens"

_jcm, 2024, doi:10.3390/jcm13102776_

Round 1
Reviewer 1 Report
Comments and Suggestions for Authors
I would like to congratulate the authors for the manuscript they present, but in my opinion it can be improved. My comments are organized below for your consideration. I hope my comments are useful for the study author(s) and editorial staff.
#Introduction: The introduction is well structured and covers the most relevant topics on the theme under study. Most of the references cited are relevant and up-to-date.
#Materials and Methods: This chapter is very well structured, as the authors describe all the procedures carried out very clearly. They demonstrate methodological rigour.
#Results: I suggest that the tables (1 / 3 / 4) be presented before the respective description of the data, as is shown in table 2.
#Discussion: The discussion is very descriptive, not very analytical. In general, is not supported by recent studies, as the vast majority of the studies referred to are over 5 years old, so I think it would be important to cite more recent studies, less than 5 years old and the discussion should be more reflective and analytical.
#The authors should update the references, as around 57% are more than 5 years old.
Author Response
Dear Reviewer,
The authors are grateful for the thoughtful analysis of our manuscript and insightful comments. We have carefully considered and applied your feedback to improve our manuscript. All revisions were highlighted using the "Track Changes" function in Microsoft Word.
Below we answer your suggestions point by point. Please note that your comments are in italics and our responses are in regular font for readability.
I would like to congratulate the authors for the manuscript they present, but in my opinion it can be improved. My comments are organized below for your consideration. I hope my comments are useful for the study author(s) and editorial staff.
#Introduction: The introduction is well structured and covers the most relevant topics on the theme under study. Most of the references cited are relevant and up-to-date.
Reply: Thank you, for the positive feedback on the Introduction section.
#Materials and Methods: This chapter is very well structured, as the authors describe all the procedures carried out very clearly. They demonstrate methodological rigour.
Reply: Thank you, for the positive feedback on the Materials and Methods section. We have tried to be precise and rigorous in describing this section.
#Results: I suggest that the tables (1 / 3 / 4) be presented before the respective description of the data, as is shown in table 2.
Reply: Thank you for this suggestion. Since Tables 1, 3, 4 are quite lengthy, we tried to put them on one page for readability. If the Tables are placed before the description of the data, they will not fit on one page and will be separated. Or there will be a lot of empty space on the pages if we leave the Tables on a full page. This will complicate the perception. It is possible that in the final version of the paper the Tables will be arranged differently. Therefore, we do not see any possibility to arrange the Tables in a different way.
#Discussion: The discussion is very descriptive, not very analytical. In general, is not supported by recent studies, as the vast majority of the studies referred to are over 5 years old, so I think it would be important to cite more recent studies, less than 5 years old and the discussion should be more reflective and analytical.
Reply: We expanded the Discussion section and tried to bring in more logical explanations of the findings.
Indeed, about half of the literature references are older than 5 years. First, this is because we have cited articles on clinical scales that were published a long time ago, but have not lost their relevance and are still in use today. Second, in some cases we found references to similar work only in older publications. Nevertheless, we conducted an additional literature search and added 15 recent articles.
#The authors should update the references, as around 57% are more than 5 years old.
Reply: We have added 15 recent articles and replaced some articles with similar more recent articles.
Thank you for your valuable suggestions to improve our manuscript.
Best regards
Authors
Reviewer 2 Report
Comments and Suggestions for Authors
This study is interesting, however there are a few factors to consider.
1- The detailed description of the measure inflammatory markers should be mentioned in the abstract as other markers.
2- ESR and CRP are neither sensitive nor specific. What are the cut-off points of CRP.
3- The sample size of AWS with DT is small and why there are a difference in the n in table 2 (22, 17)?
4- What were the CVs for the assays? Did you evaluate the performance of each ELISA kit prior to use?
5- The author focusses in the conclusion on the role of Inflammatory biomarkers in patients with DT than in patients without DT. Why did the author not measure IL-6 OR TNF-alpha?
6- What about the gender of the included patients? Could the author consider gender in the regression analysis?
7- Do any of the patients smoke? Could the author consider smoking in the regression analysis?
8- The reference needs to be updated.
9- The authors should include study limitations in the abstract.
Author Response
Dear Reviewer,
We thank the reviewer for the thoughtful evaluation of our study and valuable suggestions. We have carefully considered and applied your feedback to improve our manuscript. All revisions were highlighted using the "Track Changes" function in Microsoft Word.
Below we answer your suggestions point by point. Please note that your comments are in italics and our responses are in regular font for readability.
1- The detailed description of the measure inflammatory markers should be mentioned in the abstract as other markers.
Reply: Thank you for that suggestion. We mentioned analyzed inflammatory biomarkers in the abstract.
2- ESR and CRP are neither sensitive nor specific. What are the cut-off points of CRP.
Reply: We fully agree that ESR and CRP are non-specific markers of inflammation. According to the hsCRP kit manufacturer's manual, a concentration above 5 mg/L indicates inflammation. Concentrations of 10-30 mg/L correspond to chronic low-grade inflammation. Concentrations greater than 30 mg/L correspond to acute inflammation.
3- The sample size of AWS with DT is small and why there are a difference in the n in table 2 (22, 17)?
Reply: We agree that the sample of AWS with DT is rather small. But this is due to the relatively rare occurrence of patients with this condition. Among people hospitalized with AUD, DT develops in 3-5% of patients. In Table 2, n represents the number of patients. The value of n differs in some cases because not all patients had enough blood drawn for a complete biochemical analysis.
4- What were the CVs for the assays? Did you evaluate the performance of each ELISA kit prior to use?
Reply: Yes, for each kit we performed preliminary analyses to assess intra-assay CVs. Intra-assay CVs were consistent with those stated by the manufacturer and did not exceed 10%. We also added several identical samples to each of the plates to monitor inter-assay CVs. Inter-assay CVs for FABP2 were slightly higher than for the other kits and ranged from 12-14%, but it was within the allowed 15%. Inter-assay CVs for the other sets were less than 13%.
5- The author focusses in the conclusion on the role of Inflammatory biomarkers in patients with DT than in patients without DT. Why did the author not measure IL-6 OR TNF-alpha?
Reply: Thank you for that suggestion. Our conclusion is based on data on WBC count, PLT count, ESR and PLR. These biomarkers are non-specific biomarkers of inflammation. We did not have the opportunity to measure IL-6 or TNF-alpha due to limited blood plasma volume for analysis. In the future, we plan to study other inflammatory biomarkers.
6- What about the gender of the included patients? Could the author consider gender in the regression analysis?
Reply: All patients included in the study were male. We have indicated this in Sections 2.1. and 3.1. But for clarity, we have added a line in Table 1 on the sex of the participants. Because gender was similar in all groups, we did not include this variable in the regression analysis.
7- Do any of the patients smoke? Could the author consider smoking in the regression analysis?
Reply: Unfortunately, this information was not collected during the patient interviews. We see this as a limitation of this paper, so we have added it to the Limitations section.
8- The reference needs to be updated.
Reply: We have updated the references according to the JCM requirements. In addition, we have added more recent references.
9- The authors should include study limitations in the abstract.
Reply: The allowed number of words of the abstract is limited, so it is not possible to list all the limitations of this paper in the abstract. However, we have significantly revised the Limitations section and listed all the limitations of this paper.
Thank you for your thorough analysis of the manuscript and valuable suggestions.
Best regards
Authors
Reviewer 3 Report
Comments and Suggestions for Authors
Here are some weaknesses of each section in the article "Biochemical, Hematological, Inflammatory, and Gut Permeability Biomarkers in Patients with Alcohol Withdrawal Syndrome with and Without Delirium Tremens":
Introduction:
Ÿ The introduction could more explicitly state the study's objectives or specific
research questions, making the scope clearer to the reader.
Methods:
• While the description of the participant groups and sampling methods is comprehensive, the study's generalizability is limited by its all-male sample and the exclusion of certain demographics and comorbid conditions.
• The selection criteria and rationale for focusing exclusively on male participants could be better explained, as it may affect the applicability of the findings to broader populations.
Results:
• Some results, particularly regarding gut permeability biomarkers, are mentioned to have no significant difference, which could be discussed more critically in terms of study power or the implications for future research.
Discussion:
• The limitations are mentioned, but a more comprehensive discussion on the impact of these limitations on the study's conclusions and potential ways to address them in future research would be beneficial.
Here are some constructive suggestions for the authors
1. Expanding Participant Diversity: Consider including a more diverse participant pool in future studies to enhance the generalizability of the findings. If possible, include female participants or explain the rationale for focusing solely on male participants in greater detail.
2. Methodological Clarifications: Provide more details about the exclusion criteria and their justification to help readers understand potential biases or limitations introduced by these choices. For instance, why were patients with certain comorbid conditions excluded, and what impact might this have on the applicability of the results?
3. In-Depth Discussion of Non-Significant Findings: Address the non-significant findings, especially in the context of gut permeability biomarkers, more critically. Discuss potential reasons why these expected differences were not observed and suggest how future studies might better explore these aspects.
4. Addressing Limitations More Comprehensively: Provide a more detailed discussion of the study's limitations, including how they might impact the conclusions drawn. Suggest ways in which these limitations could be addressed in future research to strengthen the validity and reliability of the findings.
By addressing these suggestions, the authors could enhance the robustness and impact of their study, thereby contributing more effectively to the understanding and management of alcohol withdrawal syndrome and delirium tremens.
Author Response
Dear Reviewer,
The authors are grateful for the thoughtful analysis of our manuscript and insightful comments. We have carefully considered and applied your feedback to improve our manuscript. We believe that these changes have improved our paper and clarified our data presentation. All revisions were highlighted using the "Track Changes" function in Microsoft Word.
Below we answer your suggestions point by point. Please note that your comments are in italics and our responses are in regular font for readability.
Here are some weaknesses of each section in the article "Biochemical, Hematological, Inflammatory, and Gut Permeability Biomarkers in Patients with Alcohol Withdrawal Syndrome with and Without Delirium Tremens":
Introduction:
- The introduction could more explicitly state the study's objectives or specific research questions, making the scope clearer to the reader.
Reply: Thank you for this suggestion. The main aim of this work was to investigate biochemical, hematological, inflammatory and gut permeability biomarkers in AWS patients depending on the presence of DT and in comparison with healthy controls. In this work it was planned to identify biomarkers characteristic of the more severe course of AWS, i.e. delirium. The working hypothesis was that the pathogenesis of AWS and DT is partly related to inflammation caused by increased gut permeability. Part of the hypothesis of excess gut permeability in DT was not confirmed. We have added this information to the introduction.
Methods:
- While the description of the participant groups and sampling methods is comprehensive, the study's generalizability is limited by its all-male sample and the exclusion of certain demographics and comorbid conditions.
- The selection criteria and rationale for focusing exclusively on male participants could be better explained, as it may affect the applicability of the findings to broader populations.
Reply: Indeed, only males were included in this study. This is because males are more likely to be hospitalized with AWS. In addition, the inclusion of patients took place in the male department where only men are admitted, so we were not able to recruit women for the study. We are aware that all-male sample is a limitation of the study, so we have indicated this in the Limitations section.
The exclusion criteria for some somatic pathologies are explained by the fact that these diseases can strongly affect the analyzed parameters. First of all, many of these pathologies (autoimmune diseases, cancer, acute infectious, acute allergic reactions, HIV infection or viral hepatitis) induce an inflammatory response, which may affect the level of hematologic, biochemical and inflammatory biomarkers. Patients with alcoholic hallucinosis, alcoholic delusional psychosis, and alcoholic encephalopathy were also excluded because they were few and we were unable to recruit a group to analyze them separately. In addition, we excluded patients with other addictions in order to focus only on the effect of chronic alcohol abuse on the indicators studied.
We have briefly presented this information in Section 2.1.
Results:
- Some results, particularly regarding gut permeability biomarkers, are mentioned to have no significant difference, which could be discussed more critically in terms of study power or the implications for future research.
Reply: Thank you for this suggestion. We have added more discussion of the results on permeability biomarkers in section 4.3. We mentioned the limitations of the sample used and the need for replicative studies on a larger sample of AWS patients especially with DT. In addition, we reviewed the dynamic nature of changes in gut permeability biomarkers in AWS and the need for longitudinal studies.
Discussion:
- The limitations are mentioned, but a more comprehensive discussion on the impact of these limitations on the study's conclusions and potential ways to address them in future research would be beneficial.
Reply: Thank you for this suggestion. Indeed, any research has limitations and it is important to disclose this information. We have significantly revised the Limitations section and proposed potential ways to address these limitations in future research. Please see the Limitations section.
Here are some constructive suggestions for the authors
- Expanding Participant Diversity: Consider including a more diverse participant pool in future studies to enhance the generalizability of the findings. If possible, include female participants or explain the rationale for focusing solely on male participants in greater detail.
Reply: Thank you for this suggestion. Of course, in future studies we will try to include both men and women in the study. We have already described the reasons for including only men in this study earlier.
- Methodological Clarifications: Provide more details about the exclusion criteria and their justification to help readers understand potential biases or limitations introduced by these choices. For instance, why were patients with certain comorbid conditions excluded, and what impact might this have on the applicability of the results?
Reply: We have added a description of the reasons for excluding some comorbid conditions in section “2.1. Participants” and explained it in detail earlier.
- In-Depth Discussion of Non-Significant Findings: Address the non-significant findings, especially in the context of gut permeability biomarkers, more critically. Discuss potential reasons why these expected differences were not observed and suggest how future studies might better explore these aspects.
Reply: We have added more discussion of the results on permeability biomarkers in section 4.3. We indicated that the lack of significant differences may be caused by insufficient sample power and suggested future studies that would help to better understand the changes in the analyzed biomarkers.
- Addressing Limitations More Comprehensively: Provide a more detailed discussion of the study's limitations, including how they might impact the conclusions drawn. Suggest ways in which these limitations could be addressed in future research to strengthen the validity and reliability of the findings.
Reply: We have significantly revised the Limitations section and proposed potential ways to address these limitations in future research.
By addressing these suggestions, the authors could enhance the robustness and impact of their study, thereby contributing more effectively to the understanding and management of alcohol withdrawal syndrome and delirium tremens.
Thank you for your thorough analysis of our manuscript and constructive suggestions. We believe that all of your suggestions were helpful.
Best regards
Authors
Round 2
Reviewer 2 Report
Comments and Suggestions for Authors
The authors address my comments